# Different Responses of Bacteria and Archaea to Environmental Variables in Brines of the Mahai Potash Mine, Qinghai-Tibet Plateau

**DOI:** 10.3390/microorganisms11082002

**Published:** 2023-08-03

**Authors:** Linglu Xie, Shan Yu, Xindi Lu, Siwei Liu, Yukai Tang, Hailong Lu

**Affiliations:** 1School of Earth and Space Sciences, Peking University, Beijing 100871, China; xielinglu@yeah.net (L.X.);; 2Beijing International Center for Gas Hydrate, School of Earth and Space Sciences, Peking University, Beijing 100871, China; 3National Engineering Research Center for Gas Hydrate Exploration and Development, Guangzhou 511466, China

**Keywords:** microbial diversity, microbial response, potash mine, brine hydrochemistry, halophilic methanogens, Illumina Novaseq sequencing

## Abstract

Salt mines feature both autochthonous and allochthonous microbial communities introduced by industrialization. It is important to generate the information on the diversity of the microbial communities present in the salt mines and how they are shaped by the environment representing ecological diversification. Brine from Mahai potash mine (Qianghai, China), an extreme hypersaline environment, is used to produce potash salts for hundreds of millions of people. However, halophiles preserved in this niche during deposition are still unknown. In this study, using high-throughput 16S rRNA gene amplicon sequencing and estimation of physicochemical variables, we examined brine samples collected from locations with the gradient of industrial activity intensity and discrete hydrochemical compositions in the Mahai potash mine. Our findings revealed a highly diverse bacterial community, mainly composed of *Pseudomonadota* in the hypersaline brines from the industrial area, whereas in the natural brine collected from the upstream Mahai salt lake, most of the 16S rRNA gene reads were assigned to *Bacteroidota*. *Halobacteria* and halophilic methanogens dominated archaeal populations. Furthermore, we discovered that in the Mahai potash mining area, bacterial communities tended to respond to anthropogenic influences. In contrast, archaeal diversity and compositions were primarily shaped by the chemical properties of the hypersaline brines. Conspicuously, distinct methanogenic communities were discovered in sets of samples with varying ionic compositions, indicating their strong sensitivity to the brine hydrochemical alterations. Our findings provide the first taxonomic snapshot of microbial communities from the Mahai potash mine and reveal the different responses of bacteria and archaea to environmental variations in this high-altitude aquatic ecosystem.

## 1. Introduction

Hypersaline systems, such as salterns, evaporite deposits and salt lakes, are harsh environments with salt concentrations that are substantially higher than the salinity of seawater, generally approaching or exceeding salt saturation [1]. Diverse groups of halophilic microorganisms, denoted halophiles, survive or even thrive in these ecosystems [2], and these organisms exist in all three domains of life: Archaea, Bacteria and Eukarya [3]. The class *Halobacteria* is comprised of a large and phenotypically diversified group of halophilic archaea within the phylum *Halobacterota* [4]. They are the most salt-resistant and salt-requiring microorganisms within the Archaea domain and need more than 150 g/L NaCl for development and structural integrity [2]. A few species of methanogenic Archaea, the majority of which are members of the family *Methanosarcinaceae*, have evolved to survive in environments with high salt concentrations [5]. Recently, novel archaeal lineages, such as the nanosized *Nanohaloarchaea* and the methanogenic *Methanonatronarchaeia*, were discovered in hypersaline settings [6,7]. In addition, a wide variety of halophilic and halotolerant microorganisms are discovered in the Bacteria domain, which is separated into many phylogenetic subgroups [8]. Although halophiles are rare in the Eukarya domain, the green alga *Dunaliella* is commonly found in high-salt settings [9].

The investigation of microorganisms in hypersaline settings has garnered significant interest from scientists. The majority of studies on the microbial ecology of hypersaline habitats have concentrated on saline soils, saline lakes and solar salterns [6,10,11,12,13,14,15]. As an important representative of the hypersaline environment, microbial patterns within the high-altitude salt mine aquatic ecosystems have received less attention [16]. High-altitude salt mine ecosystems are (poly)extreme environments influenced heavily by high salt concentrations, high daily temperature changes, high solar UV radiation and low air pressure. They are also highly heterogeneous habitats with environmental fluctuations, including physiochemical changes, and the different intensities of anthropogenic activities resulting from the mining process [17].

As a large agricultural country, China is one of the countries with the highest potash fertilizer demand in the world. However, China’s total potash resources only account for 1.8% of the global potassium reserve [18]. These reserves are mainly distributed in western provinces, including Qinghai, Xinjiang and Tibet [19]. In the potash industry, solution mining techniques are widely used to extract minerals from potassium-bearing deposits for the exploitation and utilization of low-grade solid potash ore [20]. The deep potash salts are dissolved and extracted with water for further refining. Such methods generate substantial amounts of salt-affected fluids (brines) [21], creating hypersaline settings where only salt-tolerant microorganisms can survive [17].

Mahai potash mine is a modern potash deposit located in China’s Qinghai-Tibet Plateau. In the pressure brine layer, where the brine is predominantly chemical inter-crystalline pressurized groundwater, a considerable amount of potash resources is distributed [22]. During the solution mining operation, there are significant fluctuations in both the seepage field and the chemical field, disturbing the equilibrium and causing changes to the chemical composition and dynamics of the brine [23]. Some studies have investigated the geological, sedimentological and hydrochemical aspects of the Mahai potash mine, considered as a setting with changing environmental factors [24,25,26]. However, the microbial communities inhabiting the Mahai potash mine have not yet been explored. It is important to acquire a comprehensive phylogenetic information of the microbial communities present in the Mahai potash mine and to better understand how microbial communities are structured within this ecosystem. In addition, given that microorganisms discovered in the potash brine have the potential to be utilized in the removal of metals from groundwater contaminated by industrial activities as metal adsorbents, investigations of microbial community compositions and their associations with physicochemical variations are necessary for the development of microbially based bioremediation methods [27]. The first objective of the present study was to identify prokaryotic populations (Bacteria, Archaea) of Mahai potash mine brine using culture-independent tools targeting 16S rRNA. The second was to understand how microbial communities are shaped by the human activity and dynamic hydrochemisty of the brine within this ecosystem. In this study, high-throughput sequencing and geochemical analyses are combined to analyze the microbial diversity of the brine from the Mahai potash mine. In situ microbiome relates to brine ionic compositions, and the intensity of industrial activities are identified and inferred. Our results on microbial taxonomic composition and distribution profiles reported in this study provide the first characterization of the microbial community occurring in this particular hypersaline habitat and emphasize the different adaptive features of bacteria and archaea with environmental variations, expanding our knowledge on microbial ecology in hypersaline environments.

## 2. Materials and Methods

### 2.1. Sites Descriptions and Sampling

Mahai Salt Lake is a flat plain salt lake situated at an elevation ranging from 2743 m to 2750 m. It is located at geographical coordinates of 94°03′ E to 94°19′ E longitude and 38°02′ N to 38°35′ N latitude. The lake contains abundant potassium, magnesium and sodium component elements, and it is also the water source of the Mahai potash mine.

This study was conducted in the Mahai Salt Lake potash mine, located in the northeastern part of the inland Qaidam Basin in the Qinghai-Tibet plateau, China. The climate of this area is classified as arid continental, and it boasts an average annual rainfall of just 29.61 mm and average potential evaporation rates of 3040 mm y^−1^ [22,28]. Both solid and liquid potassium resources are found in the Mahai potash mine, which starts at the front of Saishiteng Mountain in the north, extends to ChaKa Lake and South Baxian in the south, Lenghu in the northwest and Mahai Lake in Dezong in the east, covering an area of about 3700 km^2^. The solid potassium in the Mahai potash area has been completely exploited, and the pressure brine obtained by dissolving the low-grade solid potassium rock has become the most crucial potassium resource.

In December 2020, a total of 24 brine samples (300 mL each) were collected from eight sampling sites in the Mahai potash mine area (Figure 1). Based on the intensity of the industrial activities, these sampling sites can be divided into three groups: (1) Natural site (ML): Mahai Lake located in the southeast of the mining area is the water supply for the mining process in the Mahai potash area. One surface brine sample ML was collected from the Mahai Lake and taken as the natural, original sample for comparison; (2) Monitoring sites (MS): the sub-ground brine of the potash mine was kept pumping out here for monitoring the hydrochemical variations during mining processes. Two deep brine samples were collected from monitoring stations MSa and MSb, which were considered subject to moderate human interference; (3) Mining sites (MW): 21 brine samples collected from different depths of the five mining wells (MWa, MWb, MWc, MWd and MWe) in the industrial area were used as sample counterparts under severe impacts from industrial activities (Table 1) (Appendix A). The distribution of sample numbers from the given sites is as follows: one water sample from the ML location, one water sample each from MSa and MSb locations, four water samples from different depths at the MWa location, four water samples from different depths at the MWb location, five water samples from different depths at the MWc location and four water samples from different depths at the MWd location.

ML and MS samples were collected directly by sterile polypropylene bottles, and the brines from the mining wells were collected using a depth-keeping sampling barrel. All samples were transported with ice packs in a cryogenic chamber, and the formation of salt precipitates in the ML and MW samples was observed after reaching the laboratory, where 500 mL of each brine sample was immediately filtered through 0.22 μm pore-size polycarbonate membranes (Millipore, Burlington, MA, USA) to retain prokaryotic cells. The membranes were then stored at −80 °C before DNA extraction.

### 2.2. Physicochemical Characterization

Conductivity and pH were measured with a YSI Professional Series Plus multiparameter. Salinity was measured in triplicates with a refractometer on spectrophotometer 1/100 dilutions in Milli-Q water. Brine samples used for ion composition analysis were first filtered through 0.22 µm pore-size needle-type filters (Millipore). Anionic and cationic compositions of filtered brines were determined using a Dionex ICS-2000 ion chromatograph and a Dionex Integraion HPIC ion chromatography (Dionex, Sunnyvale, CA, USA) [29,30].

### 2.3. DNA Extraction, PCR and 16S rRNA Gene Sequencing

All filtered membranes were commissioned to Novogene Technology Co., Ltd. (Tianjin, China) for DNA extraction. The CTAB method was used to extract the total sample DNA [31], after which the purity and concentration of the DNA were examined on agarose gel electrophoresis (AGE) and by using nanodrop (Thermo Scientific, Waltham, MA, USA). Then, bacterial and archaeal 16S rRNA genes V4–V5 region were amplified from the diluted genomic DNA using primer pairs with unique barcodes (Appendix A) [32,33]. Polymerase chain reactions (PCR) were performed in replicates with Phusion^®^ High-Fidelity PCR Master Mix (New England Biolabs, Ipswich, MA, USA). Each 30 µL PCR reaction contained 15 µL Phusion Master Mix (2×), 1 µM PrimerF, 1 µM PrimerR and 10 ng gDNA with ddH_2_O added to bring the final volume to 30 μL. PCRs were performed under the following conditions: 98 °C for 1 min; 30 cycles including 98 °C, 10 s; 50 °C, 30 s; 72 °C, 30 s; 72 °C, 5 min. Negative control of DNA extraction was performed in parallel by the same protocol without adding brine samples and showed a negative PCR result. The PCR products from triplicate samples were purified by magnetic beads (Qiagen, Valencia, CA, USA), quantified by enzyme labeling and mixed in equal amounts according to the concentration of PCR products. After fully mixing, the PCR products were detected by 2% agarose gel electrophoresis, and the target bands were recovered using the Universal DNA Purification Kit (TianGen, Beijing, China). The next-generation sequencing was performed by an Illumina Novaseq 6000 platform (Novogene, Beijing, China) to generate 250 bp pair-end reads with an average sequencing depth of ten million reads per sample.

### 2.4. Sequence Processing and Bioinformatic and Statistical Analysis

The acquired 16S rRNA gene sequences were processed using EasyAmplicon v1.0 according to Liu et al. [34] in R 4.0.3 [32]. The raw amplicon reads were merged; barcodes were removed; and quality filtering and singleton removing were conducted using subcommand fastq_mergepairs, fastq_stripleft (stripright), fastq_maxee_rate and minuniquesize, respectively, in VSEARCH (version 2.14.1) [33].

Chimeric sequences were eliminated using command -unoise3 in USEARCH (version 10.2.240) [35] and command uchime_ref in VSEARCH (version 4.2) [36]. ASVs table were subsequently generated using the command usearch_global in VSEARCH. The representative sequence of each ASV was assigned to a taxonomic lineage by classification against the SILVA database 138.1 [37] for bacterial and archaeal 16S rRNA genes, respectively. The non-bacterial and non-archaeal sequences were removed from the analyses based on the taxonomic classification output.

Alpha diversity indexes including Good’s coverage, Richness (observed number of ASVs), Chao1, Shannon and Inverse Simpson were calculated and visualized using the R package vegan (version 2.6.4) [38] and package amplicon (version 1.14.2) [39], respectively. Rarefaction curves were also visualized using the package amplicon. Constrained principal coordinate analysis (CPCoA) was firstly performed using USEARCH to generate an Unweighted UniFrac distance matrix file. Subsequently, the dissimilarities among sample groups were assessed for significance using the Anosim test, and the results were visualized by the package ggplot2 (version 3.3.6) [40]. The taxonomic composition of phylum level of each sample group was also visualized by package ggplot2. Venn diagrams were plotted by the package ggVennDiagram (version 1.2.2) [41]. Pearson’s correlations between the environmental factors and the alpha-diversity indexes or microbial species were calculated using the “cor” function in package psych (version 2.2.9) [42] and were plotted using package pheatmap (version 1.0.12) [43]. Principal Component Analysis (PCA) plot of samples according to their ionic composition was visualized by the “fviz_pca_var” function in package factoextra (version 1.0.7) [44]. Redundancy analysis (RDA) was performed using Canoco 5.0 software [45] to analyze the correlation between the methanogenic community and ionic concentrations, and significant combinations were selected.

## 3. Results

### 3.1. Physiochemical Characterization of Brine Samples of Mahai Lake Potash Mine

The lowest salinity (1.5%, *w*/*i*) and highest pH value (7.95) was found in the natural lake water sample ML. By contrast, MS and MW samples collected from the industrial area exhibited much higher salinity ranging from 9.8–14.7% (*w*/*v*). They were pH-neutral (6.0–7.4), NaCl-dominated hypersaline brines that were also enriched in other cations and anions such as K^+^, Mg^2+^, Ca^2+^, SO_4_^2−^, NO^3−^ and F^−^. In the five mining wells (MWa, MWb, MWc, MWd and MWe), brine salinity marginally increased with the sampling depth while the pH value decreased. The measured physicochemical parameters of all brine samples were summarized in Table 1 and Appendix A.

To determine the potential relationships between local chemistry and microbial communities, we first applied a Principal Component Analysis (PCA) to classify the brine samples based on the ionic composition (Appendix A). The first two principal component axes in the PCA model explain 66.8% of the total data variance, and the PCA plot combined with variation analysis of the ionic composition of brine samples identified four main brine types from all 24 brine samples (Figure 2A): *Brine type ML* with a negative PCA1 score and a slightly positive PCA2 score was characterized by low Mg^2+^, K^+^, Ca^2+^ and Cl^−^ concentrations; *Brine type MS* with positive PCA2 scores could be characterized compositionally by high SO_4_^2−^ and F^−^ concentrations. In addition, the other two brine types MWace and MWbd, separated by axis 2, were characterized by medium to high concentrations of Mg^2+^, K^+^, Ca^2+^ and high concentrations of Na^+^, respectively (Figure 2A, Appendix A). These results indicated the hydrochemical diversity of brines in this heterogeneous hypersaline system.

### 3.2. Alpha Diversity of the Prokaryotic Communities within the Mahai Potash Mine Area

Despite numerous attempts, we failed to obtain the amplicons with archaea-specific primers from samples MWa1 and MWc3. After the quality filtration, denoising and removal of singletons, we obtained 1,789,850 reads and 3980 ASVs for bacteria and 159,712 reads and 238 ASVs for archaea by sequencing amplicon pools using Illumina NovaSeq (Appendix A). The constructed rarefaction curves indicated that the sequencing depths were adequate to cover the prokaryotic diversity in all collected samples (Appendix A).

For alpha diversity, a much higher diversity of bacteria was detected, compared to that of the archaeal domain (Figure 3A, Appendix A). Notably, the assessment of the diversity index Chao1 showed differences in the complexity of archaeal communities among hypersaline brine samples collected from mining wells (MW samples) in the industrial area. Boxplot based on ASV level showed that the Chao 1 indexes were significantly different among the mining wells, showing the highest diversity in mining wells MWa and MWc, and the lowest in MWe (Figure 3A, right). Nevertheless, the Chao1 index of bacteria had no significant difference among the sampling sites (Figure 3A, left). The results indicated that the diversity of archaeal communities was more sensitive to the environmental variations of the hypersaline brines.

Correlations of prokaryotic diversity indexes and physicochemical parameters of the brine samples were further analyzed. In terms of bacteria, the results of the coefficient of correlation (r) showed a significant negative correlation between equitability and brine conductivity (*r* = −0.45; *p* = 0.047) (Figure 3B, left). For the archaeal community, significant negative associations of K^+^ concentration and buzas_gibson (*r* = −0.61; *p* = 0.007), NO_3_^−^ concentration and inversimpsion (*r* = 0.56; *p* = 0.01), Shannon index (*r* = 0.54; *p* = 0.02) and jost1 index (*r* = −0.54; *p* = 0.02), Cl^−^ concentration and equitability (*r* = 0.52; *p* = 0.027) were detected. NO_3_^−^ concentration was significantly positively correlated with the berger_paker index, which reflects the frequency of the most abundant ASVs (*r* = 0.55; *p* = 0.02) (Figure 3B, right). Notably, although not significant, opposite trends of bacterial and several archaeal diversity indexes (such as Shannon, ACE, Chao 1 and Richness indexes) related to environmental parameters were observed: bacterial diversity rose with increasing depth, Ca^2+^ and Mg^2+^ concentration and dropped with elevated concentration of Na^+^, whereas archaeal showed opposite relationships with these parameters (Figure 3B). In contrast, the anions showed consistent relations with diversity indexes of archaea and bacteria. These associations indicated that the selection effect of these cations on archaeal and bacterial communities may be diametrically opposite.

### 3.3. Characterization of the Prokaryotic Communities

Based on the bacterial taxonomic composition, a total of 51 phyla were identified (Figure 4A). Collectively, *Pseudomonadota* (former phylum *Proteobacteria*, with gamma and alpha subdivisions) was the most prevalent phylum detected in the Mahai potash mine, accounting for 78.87% of the total number of bacterial sequences. To a lesser extent, *Bacteroidota* (7.41%), *Bacillota* (5.27%), *Actinomycetota* (4.20%), *Desulfobacterota* (0.90%) and *Chloroflexota* (0.59%) were also present (Figure 4A). *Bacteroidota* (*Flavobacteriales*) was the most dominant phylum in the natural sample ML, representing 72.9% of its bacteria community (Figure 4 and Appendix A). The phyla composition of the bacterial domain detected in brine samples from the industrial area was similar. *Gammaproteobacteria* took precedence in MS and MW sites (59.35%), followed by *Alphaproteobacteria*, accounting for 11.20% to 32.33% of all read counts within each sample. Furthermore, we detected an increased abundance of ASVs affiliated with *Alphaproteobacteria* in MW sites (Figure 4A). In order level, in addition to *Flavobacteriales*, sample ML possessed highly abundant *Pseudomonadales* whereas *Burkholderiales* and *Rhizobiales* dominated other hypersaline brine samples from the industrial area (Appendix A).

Altogether, eight archaeal phyla were identified by amplicon sequencing from brine samples of the Mahai potash mine, mainly dominated by *Halobacterota*, *Euryarchaeota*, *Thermoproteota* and *Nanohaloarchaeota*, contributing up to 96.70% of the total archaeal sequences. *Thermoplasmatota*, *Micrarchaeota*, *Nanoarchaeota* and *Hadarchaeota* also presented in low abundance (Figure 4B). Different from the distribution of bacteria, the archaea composition in ML and MS sites was more similar, compared to MW sites. Taxonomic analysis based on relative abundance revealed that *Halobacterota* was the most predominant phyla (above 66.3%) in ML and MS samples. In contrast, MW samples only had an average abundance of 31%, they were characterized by higher relative abundances of *Euryarchaeota* (20.9–51.1%).

Particularly, the archaeal community in the brine samples of the Mahai potash area was highly dominated by members of the order *Halobacterales* and halophilic methanogens, together making up 63.49% of the total archaeal sequences (Figure 4C). The abundance of *Halobacterales* was significantly higher in the ML and MS sites (66.30–90.20%) compared with MW sites (3.36–56.92%). Furthermore, there were significant differences in the composition of *Halobacterales* at the genus level across these three site groups. *Halonotius, Halolamina* and *Natronomonas* were the three most abundant genera within ML and MS samples while the compositions of *Halobacterales* in the five mining wells showed no common features (Figure 4C).

Methanogenic communities in the Mahai potash mine were dominated by orders of *Methanosarcinales, Methanobacteriales* and *Methanococcales*. At a higher phylogenetic resolution, *Methanothermobacter* was the only methanogenic genus discovered in the natural sample ML (*brine type ML*) collected from the Mahai salt lake, with a relative abundance of 3.85% in the archaea domain. By comparison, samples collected from the mining wells (MWs) contained eight different methanogenic genera (*Methanothermobacter*, *Methanobrevibacter, Methanobacterium*, *Candidatus* Methanomethylicus, *Methanosaeta*, *Methanosarcina* and uncultured *Methanomassiliicoccaceae*) with higher abundances, accounting for 24.4% to 67.1% of the total archaeal sequences in each sample (Figure 4D). Specifically, methanogens obtained from mining wells MWa, MWc and MWe (*brine type MWace*) were dominant by *Methanothermobacter* (14.4% to 42.9%), while MWb and MWd (*brine type MWbd*) were dominant by *Methanobrevibacter* and *Methanobacterium* (together accounting for 23.7% and 35.9% in each archaeal community). Surprisingly, no methanogen-related sequences were recovered from the two monitoring stations (*brine type MS*).

### 3.4. Microbial Community Comparison among Sample Groups

To compare the prokaryotic community structure among different sample groups, CPCoA with unweighted unifrac distance followed by a permutation-based ANOVA (PERMANOVA, Table 2) was performed. A clear separation of bacterial communities was observed among ML, MS and MW groups (*p* = 0.021). However, there was no significant difference in archaeal compositions in these three groups (Figure 5A,E). The Venn diagram further reflected this difference. Sample groups ML, MS and MW had 469 bacterial ASVs (11.8% of total bacterial ASVs) in common and a large number of ASVs were found exclusively in the MW group (2263 ASVs, 56.9% of MW reads), in terms of archaeal community, the highest number of unique ASVs was also found in the MW group (157 ASVs, 66.0% of MW reads) (Figure 5B,F), indicating the potential peculiarities of MW habitats. Thus, we further compared the prokaryotic communities from the five mining wells (MWa, MWb, MWc, MWd and MWe). Interestingly, our results revealed that the composition of archaeal communities was significantly different (*p* = 0.009) whereas bacterial communities showed no statistically significant difference within the MW groups (*p* = 0.2) (Figure 5C,G). In addition, five mining wells had 692 bacterial ASVs in common, accounting for 17.47% of all the bacterial ASVs obtained in total. And unique ASVs, which were only found in a particular bacterial community, corresponded to 0.48% to 2.6% of the bacterial ASVs in each mining well. In comparison, only 12 ASVs affiliated with the archaeal domain (5.11% of the total archaeal ASVs) were shared by the five mining wells (Figure 5D,H). A high proportion of well-exclusive ASVs was observed in mining well MWa, MWc and MWe (4.68% to 9.79%), which constituted the *brine type MWace* and separated from *brine type MWbd* along axis 1 on the CPCoA plot (Figure 5G,H). The low proportion of shared ASVs and the high number of well-exclusive ASVs suggested that the habitat heterogenicity of the mining wells strongly shaped the archaeal community. This observation was consistent with the results of the CPCoA. Furthermore, PERMANOVA analyses on the ionic and microbial compositions showed that archaea, particularly anaerobic methanogens, shifted significantly with distinct brine hydrochemistry. Though the bacterial communities also differed among the four brine types, no significant differences were observed in the brine groups with distinct ionic compositions collected from the industrial area with high salinity (MS and MW samples) (Table 2). Our results suggested that halophilic methanogens in the Mahai potash mine were sensitive and may be responsive to the ionic changes in this hypersaline aquatic ecosystem.

### 3.5. Potential Correlations between Microbial Communities and Variables of Brines

To clarify the relationship between prokaryotic composition and environmental factors, Pearson’s correlation test was performed (Appendix A). *Desulfurococcales*, *Thermococcales*, *Woesearchaeales* and *Methanomicrobiales* were positively correlated with K^+^ while there was no significant positive correlation between Na^+^, Ca^2+^ and any archaeal genus. In comparison, no bacterial genus showed a significant correlation with K^+^, whereas multiple genera, such as *Thermoleophilia*, *Desulfuromonadia*, *Phycisphaerae*, *Synergistia*, *Calditrichia*, *Holophagae*, MB-A2-108 and *Methylomirabilia*, had significant positive relations with Mg^2+^. By using correlation analysis (RDA), the relationships between the methanogenic community and ionic concentrations were further explored. Most obviously, *Candidatus* Methanomethylicus and *Methanothermobacter* were positively correlated with K^+^, Ca^2+^ and Mg^2+^ while *Methanobacterium* and *Methanobrevibacter* showed a positive correlation with Na^+^ (Figure 2B). These results suggested different adaptations of archaea and bacteria to the hypersaline brines with distinct cationic compositions in the Mahai potash mine.

## 4. Discussion

In the conventional potash mining industry, large amounts of hypersaline brines are produced. Although similar hypersaline ecosystems can also be found in salt lakes and brines of oil-field, potash brines are thought to contain more limited carbon sources [46,47]. Despite playing a crucial biological role, the microbial inhabitants in the potash mining industry are often disregarded and rarely reported. In this study, brines from eight sampling sites of the Mahai potash mine (Qinghai-Tibet plateau, China), a heterogenic hypersaline environment, were collected to investigate the halophiles preserved in this niche and their responses to environmental factors. Based on the results of the physicochemical and microbial diversity analyses, bacterial and archaeal communities in 24 brine samples showed different distributions and responses to environmental variables, such as industrial influences and hydrochemistry.

### 4.1. Prokaryotic Compositions in the Mahai Potash Area

Overall, *Pseudomonadota* and *Bacteroidota* made up the majority of the bacterial communities in the hypersaline brine samples from the Mahai potash mining area; this observation was consistent with the findings from hypersaline salterns and concentrated brine of spa [48,49,50]. *Pseudomonadota* and *Bacteroidota* were considered dominant taxa that play a significant role in the cycling of carbon and nitrogen in hypersaline environments [48,49,51]. Detailed analysis of the bacterial community composition at the genus level showed that *Ralstonia* species (45.05% in total) and *Methylobacterium–Methylorubrum* (19.47% in total) prevailed in the industrial area while *Flavobacterium* (3.38%) was the most abundant genera in natural sample ML. The domination of class *Flavobacteria* at low salinity has been reported in the salt ponds in hypersaline Lake Meyghan [52]. Moreover, the occurrence of *Ralstonia* species has also been reported in other hypersaline environments represented by hypersaline oil reservoirs [53,54]. Another abundant bacterial genus *Methylobacterium–Methylorubrum* is facultatively methylotrophic bacteria; they can grow on methanol as the sole carbon and energy source, and several strains can also utilize methylamine [55]. In the photovoltaic panel ecosystem, *Methylobacterium–Methylorubrum* was reported to be involved in all metabolisms related to stress resistance and can tolerate low nutrient conditions [56].

Numerous haloarchaeal species, including *Arhodomonas* sp., *Haloarchaeum* sp. and *Haloferax* sp., have been discovered from a flotation enrichment step and mineral samples from a Russian potassium mining company [57,58,59]. And *Halobacterium* sp., *Haloarcula* sp. and *Halococcus* sp. were detected from mineral samples and a brine pool within a British potash mine [60]. In this study, the majority of the archaeal sequences belonged to the class *Halobacteria*. A high relative abundance of *Halobacteria* was detected in ML and MS brines, dominated by *Natronomonas*, *Halolamina*, *Halonotius* and *Halorubrum*. Previous studies suggested that *Natronmonas* is one of the most ecologically successful taxa in surviving in hypersaline environments [61,62,63,64]. *Halolamina* has been suggested to be involved in phosphorous cycle maintenance in Hypersaline ecosystems [65]. *Halonotius* is diverse and abundant in hypersaline environments; this genus may serve a Black Queen function among the haloarchaea members in hypersaline settings by *de novo* cobalamin biosynthesis [66]. Notably, among the five mining wells, *Halobacteria* composition in the genus level was different from each other, indicating their high sensitivity to the heterogeneity of hypersaline brines subjected to the severe mining process.

In hypersaline environments, methylotrophic methanogenesis is generally regarded as the predominant pathway and the order *Methanosarcinale* is responsible for this process, although there are exceptions [67,68,69]. However, in the Mahai potash mine, members of the genus *Methanothermobacter*, *Methanobrevibacter* and *Methanobacterium*, which are referred to as hydrogenotrophic methanogens, dominated the methanogenic communities. The detection and dominance of hydrogenotrophic methanogens have been reported in other hypersaline environments [70,71,72,73]. Hydrogenotrophic methanogenesis would perform in hypersaline microbial mats with the presence of certain organic compounds, such as lactate and acetate [74,75]. Further investigations are required to fully understand the potential contribution of hydrogenotrophic methanogens to the decomposition of organic matter in hypersaline environments and the syntrophic interactions between these organisms and other fermentative microbes.

### 4.2. Bacteria and Archaea Responded Differently to Environmental Factors

Since salinity impacts the physiological characteristics of cells, improves osmotic potential and decreases the activity and biomass of microorganisms, it has been reported to be an important factor that shapes the structure and composition of prokaryotic communities and affects the microbial food web in a large number of studies focused on multiple hypersaline environments [76,77,78,79,80,81,82,83,84]. In addition, previous research has demonstrated that microbial populations in terrestrial saline and hypersaline ecosystems exhibit limited taxonomic diversity [85] and that microbial diversity generally declines with rising salinity [86]. Our results suggest that salinity is not the main factor determining the diversity of the microbial community of the Mahai potash area. Analysis of the prokaryotic diversity response to environmental variables revealed that archaeal alpha diversity changed significantly among five mining wells while bacterial diversity was not sensitive to environmental changes in these hypersaline wells.

We found that anthropogenic activity and brine chemical property were the direct determinants of changes in bacteria and archaea communities, respectively. Different responses of archaeal and bacterial communities to environmental gradients have been reported in a California hypersaline lake, where salinity structured the bacterial community while archaea were weakly correlated with total carbon. A possible explanation can be ascribed to different ecological and adaptation strategies in hypersaline habitats: archaea thrive by controlling inorganic ion concentrations while halophilic bacteria synthesize specific organic molecules as compatible solutes to balance the osmotic pressure [87].

### 4.3. Archea and Brine Chemistry

In addition, we also detected that the changes in ionic compositions were responsible for the differences in methanogenic beta diversity. It has been recently reported that the composition of halophiles in salt-saturated ecosystems can vary based on ionic composition, geographical location and nutrient concentrations [61,88,89]. Some archaeal community structure differences correlated with distinct ion composition. In a biogeographical study conducted with samples from distant hypersaline environments, the community trends of total archaea, especially *Halonotius* and *Halorubellus* correlated with Mg^2+^, although Mg^2+^ is not required for the growth of *Halorubellus*. Other correlations were also observed in the abundances for specific archaeal genera and ions, such as PO_4_^3−^ ∼ *Halobonum*, F^−^ ∼ *Natronomonas* and Ca^2+^ ∼ *Halomicrobium* [90]. In addition, as reported in Salar de Uyuni, prokaryotic diversity and community structure are selected according to their location and ionic composition [91]. These findings suggest that ionic composition and chaotropicity of hypersaline brines might play an important role in the selection of some specific groups. Strain OCM283 of *Methanosarcinales* isolated from an oil-reservoir brine was reported to have a specific requirement for Ca^2+^ and Mg^2+^ [92], and methanogenesis might be influenced by increased anion concentrations in flooded soils [93]. It has been suggested that variations in ionic composition can affect nutrient availability and impact microbial respiration [94]. The ionic composition might play a significant role in selecting particular methanogenic groups, and different species have achieved distinct advantages associated with the local physiochemical characteristics.

Though the specific evolutionary processes taking place underneath in these wells have not yet been thoroughly investigated in this study, our results indicated that this system has more potential influences on methanogenic communities. While indigenous archaeal communities might exist in these environments, they also may be contaminants introduced through industrial development and subsequently selected by their high salinity tolerance.

Due to the limited number of collected sample and one-time sampling scheme, the obtained results should be treated with caution; as a consequence, the reported data cannot be considered to be representative of the complexity of the microbial community present in this area, preventing the inference of robust conclusions on microbial dynamics. Data on microbial taxonomic composition and distribution profiles related to environmental variations reported in this study add important information on the microbial diversity and changes and adaptations of halophiles in response to environmental gradients, providing new insights on this particular hypersaline aquatic habitat. Potash mine aquatic ecosystems with their heterogenic conditions are good laboratories for studying changes and adaptations of halophiles in response to environmental gradients.

## 5. Conclusions

This study provides valuable information regarding the prokaryotic communities inhabiting the hypersaline, Mahai potash mine. Bacterial and archaeal composition, diversity and their potential responses to environmental variables were investigated using modern sequencing by Illumina Novaseq and chemical analyses. In the hypersaline brines obtained from the industrial region, we discovered a diversified bacterial community dominated by *Pseudomonadota*, but in the natural brine collected from the upstream Mahai salt-lake, the majority of the 16S rRNA gene reads were assigned to *Bacteroidota*. *Halobacteria* and halophilic methanogens made up the majority of the archaeal communities. Additionally, we discovered that, in the Mahai potash mine, archaeal populations were mainly shaped by the chemical characteristics of the hypersaline brines, whereas bacterial communities tended to be affected by industrial intensities. Distinct halophilic methanogens were determined in sample groups with unique ionic compositions, demonstrating their exceptional sensitivity to hydrochemical changes in the brine. These findings provided new insights into the ecological patterns of prokaryotes in high-altitude salt mine habitats.

## Figures and Tables

**Figure 1 microorganisms-11-02002-f001:**
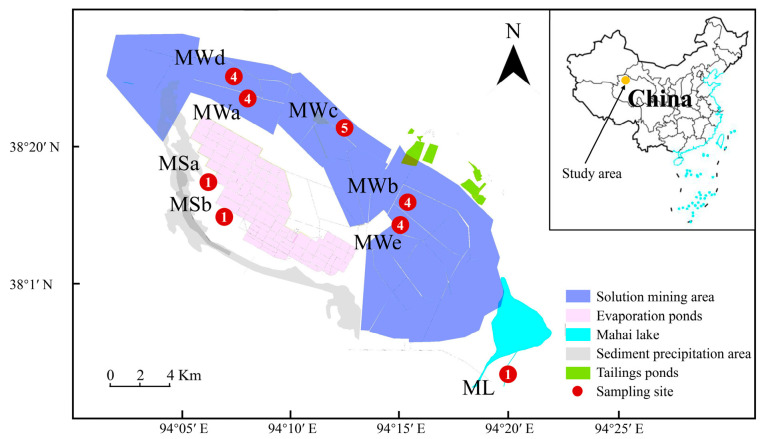
The map of the Mahai potash area in Qinghai province showing the location of the sampling sites. The eight sampling sites in this study include one site at the Mahai lake (ML) in the southeast of the mining area, two sites at the monitoring station MSa and MSb and five sites at mining wells MWa, MWb, MWc, MWd and MWe. In total, 24 brine samples were collected as indicated.

**Figure 2 microorganisms-11-02002-f002:**
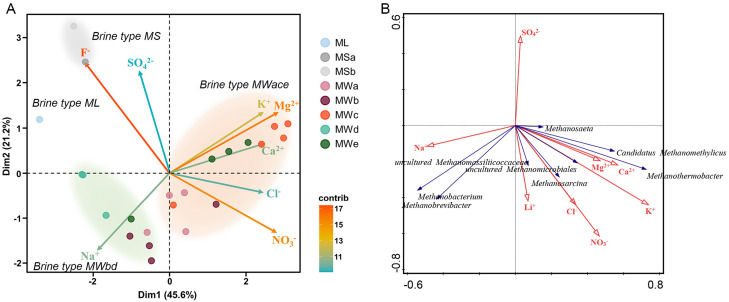
(**A**) The PCA plot of 24 brine samples of the Mahai potash region according to their ionic composition (see Appendix A). Each point represents a single sample. Colored zones highlight the groups of hypersaline brine samples corresponding to the chemical types identified in this study. (**B**) Distance−based redundancy analysis (dbRDA) of methanogenic genera in response to measured ion concentrations. *Brine type ML*: Brine type of Mahai Lake; *Brine type MS*: Brine type of Monitoring sites; *Brine type MWace*: Brine type of Mining site a, Mining site c and Mining site e; *Brine type MWbd*: Brine type of Mining site b and Mining site d.

**Figure 3 microorganisms-11-02002-f003:**
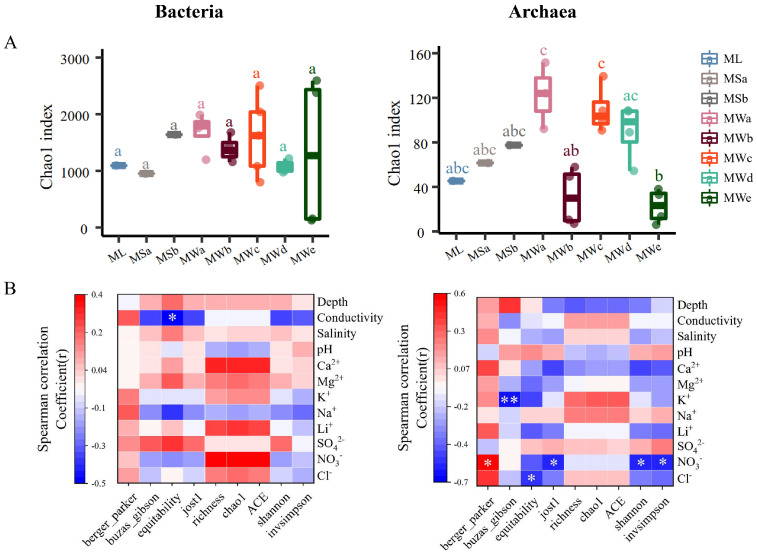
Alpha−diversity of prokaryotes of the brine samples from Mahai potash area. (**A**) Boxplots of the Chao 1 indexes of bacteria (**left** panel) and archaea (**right** panel) communities with colors indicating the sampling sites listed in the box. Boxes indicate 25th to 75th percentiles, with mean values marked as a line and whiskers indicating the minimum and maximum values. Different letters have significantly different meanings from each other (*p* < 0.05). (**B**) Heat maps demonstrate correlations between alpha−diversity indexes of bacteria (**left** panel) and (**right** panel) archaea communities and physicochemical properties of collected brines. Correlation coefficients ranged from negative to positive and are indicated by color intensity, changing from blue to red, respectively, as illustrated in the key (*, *p* < 0.05; **, *p* < 0.01).

**Figure 4 microorganisms-11-02002-f004:**
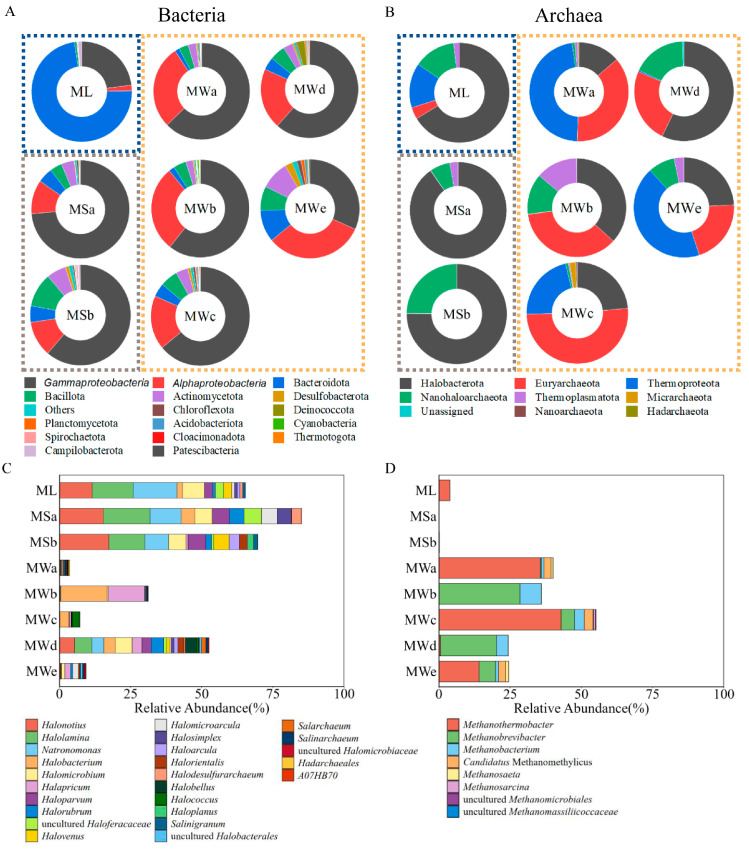
Relative proportions of the bacterial (**A**) and archaeal (**B**) phyla of the eight sampling sites illustrated in doughnut charts. *Pseudomonadota* phylum is shown at the class level. Three groups of the sampling sites ML, MS and MW are boxed up separately. Relative abundance of the class *Halobacteria* (**C**) and methanogens (**D**) are shown at the genus level.

**Figure 5 microorganisms-11-02002-f005:**
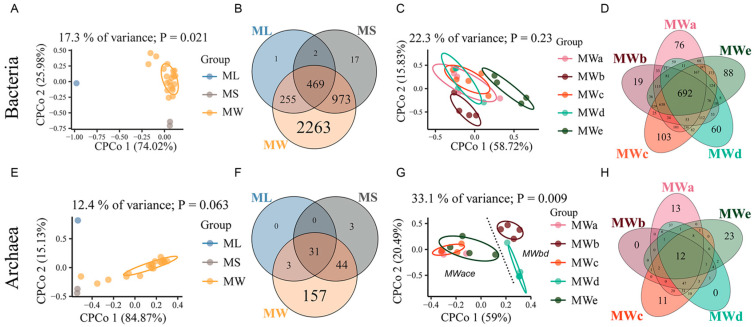
Constrained principal coordinates analyses (CPCoA) with unweighted unifrac distance at the ASV level constrained by gradient of anthropogenic activity (**A**,**E**) and five mining wells (**C**,**G**), and Venn diagrams (**B**,**D**,**F**,**H**) demonstrating shared and unique ASVs among corresponding groups. Each point represents a single community. The upper and lower panels show the results of bacteria and archaea, respectively.

**Table 1 microorganisms-11-02002-t001:** Information on the sampling sites and physicochemical parameters of brine samples.

Sampling Sites	Coordinates		Site Groups and Descriptions	Sample ID	Physicochemical Parameters			
	Lat. N	Long. E			Depth (m)	Salinity (%)	Conductivity (us/cm)	pH
**ML**	94°18′4″	38°7′59″	ML site: natural.	ML	0.2	1.50	16,928	7.95
**MSa**	94°06′57”	38°17′30″	MS sites: moderate human influence	MSa	6	11.30	202,010	7.21
**MSb**	94°06′17”	38°18′29″		MSb	11	9.80	205,610	7.37
**MWa**	94°14′2″	38°19′9″	MW sites: high human influence.	MWa1	2.3	10.20	215,260	7.09
				MWa2	3.5	9.80	n.a.	7.13
				MWa3	4.5	10.60	215,230	7.05
				MWa4	5.7	11.90	200,475	6.76
**MWb**	94°15′10″	38°17′51″		MWb1	3	10.00	240,288	7.40
				MWb2	5	10.30	241,540	7.38
				MWb3	7	10.20	240,660	7.35
				MWb4	10	11.60	213,816	6.91
**MWc**	94°12′21″	38°20′25″		MWc1	2	12.70	n.a.	6.54
				MWc2	4	13.20	190,870	6.32
				MWc3	6	13.80	185,577	6.13
				MWc4	8	14.40	179,920	6.00
				MWc5	10	14.70	176,253	6.04
**MWd**	94°07′15″	38°22′10″		MWd1	0.5	10.30	235,022	7.15
				MWd2	2	10.70	235,056	7.11
				MWd3	4	10.80	232,137	7.14
				MWd4	6	n.a.	n.a.	n.d.
**MWe**	94°14′27″	38°16′55″		MWe1	3	10.60	n.a.	7.37
				MWe2	6	12.10	187,704	6.89
				MWe3	9	12.20	166,525	6.54
				MWe4	12	14.10	163,480	6.32

ML: Mahai Lake; MS: Monitor Station; MW: Mining Well; n.a.: not acquired.

**Table 2 microorganisms-11-02002-t002:** PERMANOVA analyses of the composition of the ionic and microbial composition of brine sample groups.

Brine Groups	Ionic Composition		Bacteria		Archaea		Methanogen	
	*r*	*p*	*r*	*p*	*r*	*p*	*r*	*p*
ML/MS/MWace/MWbd	0.718	**	0.298	*	0.364	**	0.449	**
MS/MWa/MWb/MWc/MWd/MWe	0.658	**	0.275	0.19	0.454	**	0.425	**
MS/MWace/MWbd	0.443	**	0.090	0.29	0.336	**	0.425	**
MWace/MWbd	0.399	**	0.045	0.409	0.263	**	0.387	**
MWa/MWb/MWc/MWd/MWe	0.633	**	0.242	0.223	0.396	**	0.559	*

*p* <0.05, *; *p* <0.01, **.

## Data Availability

Illumina reads and accompanying metadata were uploaded to the NCBI Sequence Read Archive (SRA) under BioProject PRJNA851205.

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
