# Peer review of "Different Responses of Bacteria and Archaea to Environmental Variables in Brines of the Mahai Potash Mine, Qinghai-Tibet Plateau"

_microorganisms, 2023, doi:10.3390/microorganisms11082002_

Round 1
Reviewer 1 Report
The manuscript, entitled "Different responses of bacteria and archaea to environmental variables in brines of the Mahai potash mine, Qinghai-Tibet Plateau" submitted to Microorganisms, presents a study that used high-throughput sequencing of the 16S rRNA gene amplicon to assess changes in bacterial and archaeal communities in brine samples collected from sites of intensity of industrial activity and discrete hydrochemical compositions at the Mahai potash mine. The research is important for detecting microorganisms with potential methanogenic properties. This is especially relevant for archaeons that inhabit extreme environments, including, i.a., excessively saline areas. The article is well written, but the novel aspect of the research is not clearly indicated. In addition, the objective of the research should be detailed. Specific comments are included below:
Introduction
1. Please adjust your citation style to the requirements of the journal. Please correct in the whole manuscript.
2. Please indicate the innovative aspect of the research.
3. The objective of the research is too general and needs to be made more specific.
Materials and methods
1. Please describe the sampling in more detail (number of samples from a given site, number of replicates, etc.).
2. Lines 142-147: Please add references for methods
3. Line 170: please also add „statistical analysis” or divide this section into 2 subsection
Results
1. Line 209: Please correct names of ion in whole manuscript
2. Lines 227-231:
4. Please add an explanation of the abbreviations of the research variants
5. Please improve the quality of Figure 5
Discussion
1. Please do not cite figures and tables in the discussion section
2. Lines 420-422: I think this is not related to the subject of the manuscript and should be removed
3. Please discuss more the occurrence of archeons in the environmental conditions mentioned above
Reviewer 2 Report
Undoubtedly, the subject taken up by the authors is very interesting and important.
The authors made several significant mistakes.
1. The manuscript is not prepared in accordance with the requirements of the journal.
- incorrectly quoted literature and list.
2. line 100. 2743-2750m but what altitude? or where?
3. line 113. The authors write that the samples were taken in December 2020 in 24 places.
A one-time sampling cannot be the basis for scientific analysis, because it is subject to error. In other words, in order to draw conclusions about the human impact on the diversity of bacterial communities, samples should be taken from the same places at the same time over several years and only then should the results. The authors did not specify from what depth the samples were taken, which is an important element of the methodology. In summary, the manuscript is methodologically unpublishable.
4. Lines 280-282 it is generally accepted that Latin names should be written in italics.!!!!!
5. I do not understand why the same figures that are included in the manuscript were added to the Supplementary.
Reviewer 3 Report
This is an interesting manuscript examined the Different responses of bacteria and archaea to environmental variables in brines of the Mahai potash mine, Qinghai-Tibet Plateau. The findings contribute to a better knowledge of the ecological distribution of prokaryotes and reveal the various reactions of bacteria and archaea to environmental conditions in the high-altitude potash mine's aquatic ecosystem. This is a potentially useful interesting work, and the subject certainly falls within the general scope of Microorganisms journal. However, I have some concerns about the work presented in the MS.
1. The Abstract is not a good writing, and it should clear the significance of the study. In addition, it is better to list some data of the results.
2. The topic of this study is interesting, but the Introduction fails to reflect the highlights of this manuscript and lacks a research gap.
3. In the Background, the authors should provide rational for why study was conducted and in what aspects it is novel from previously conducted similar studies.
4. The section of Introduction, the latest literatures were cited inadequately. Here, I suggest to cite more new literatures but delete the old ones. Also, the logical relation of the research gap and this study was not well listed in the last paragraph of Background.
5. The correlations between archaeal diversity and compositions between the chemical properties of the hypersaline brines worthy of further analysis and discussion.
6. Some spelling error should be checked again.
In summary, the MS has its merits, it could be considered for publication with revision.
Round 2
Reviewer 1 Report
Thank you for responding to my comments. The manuscript now appears better and can be published in this form.
Reviewer 2 Report
Dear authors,
thank you for responding to my comments